# *Bombyx batryticatus* Protein-Rich Extract Induces Maturation of Dendritic Cells and Th1 Polarization: A Potential Immunological Adjuvant for Cancer Vaccine

**DOI:** 10.3390/molecules26020476

**Published:** 2021-01-18

**Authors:** Ha-Yeon Song, Jeong Moo Han, Eui-Hong Byun, Woo Sik Kim, Ho Seong Seo, Eui-Baek Byun

**Affiliations:** 1Research Division of Radiation Science, Advanced Radiation Technology Institute, Korea Atomic Energy Research Institute, Jeougeup 56212, Korea; songhy@kaeri.re.kr (H.-Y.S.); jmhahn@kaeri.re.kr (J.M.H.); seohoseong@kaeri.re.kr (H.S.S.); 2Department of Biotechnology, College of Life Science and Biotechnology, Korea University, Seoul 02841, Korea; 3Department of Food Science and Technology, Kongju National University, Yesan 32439, Korea; ehbyun80@kongju.ac.kr; 4Functional Biomaterial Research Center, Korea Research Institute of Bioscience and Biotechnology, Jeoupeup 56212, Korea; kws6144@kribb.re.kr

**Keywords:** *Bombyx batryticatus* extract, dendritic cells, cancer vaccine, multifunctional T cells, amino-acid composition, immunopotentiator

## Abstract

*Bombyx batryticatus*, a protein-rich edible insect, is widely used as a traditional medicine in China. Several pharmacological studies have reported the anticancer activity of *B. batryticatus* extracts; however, the capacity of *B. batryticatus* extracts as immune potentiators for increasing the efficacy of cancer immunotherapy is still unverified. In the present study, we investigated the immunomodulatory role of *B. batryticatus* protein-rich extract (BBPE) in bone marrow-derived dendritic cells (BMDCs) and DC vaccine-immunized mice. BBPE-treated BMDCs displayed characteristics of mature immune status, including high expression of surface molecules (CD80, CD86, major histocompatibility complex (MHC)-I, and MHC-II), increased production of proinflammatory cytokines (tumor necrosis factor-α and interleukin-12p70), enhanced antigen-presenting ability, and reduced endocytosis. BBPE-treated BMDCs promoted naive CD4^+^ and CD8^+^ T-cell proliferation and activation. Furthermore, BBPE/ovalbumin (OVA)-pulsed DC-immunized mice showed a stronger OVA-specific multifunctional T-cell response in CD4^+^ and CD8^+^ T cells and a stronger Th1 antibody response than mice receiving differently treated DCs, which showed the enhanced protective effect against tumor growth in E.G7 tumor-bearing mice. Our data demonstrate that BBPE can be a novel immune potentiator for a DC-based vaccine in anticancer therapy.

## 1. Introduction

Cancer immunotherapy has emerged as a powerful weapon for eradicating malignant tumors. It offers numerous advantages compared to chemotherapy, radiotherapy, and traditional surgery, such as strong efficacy against certain cancers, fewer side effects, and potential of combination therapy [1,2]. There are several classes of cancer immunotherapy, including immune checkpoint blockade therapy, chimeric antigen receptor (CAR) T-cell therapy, and cancer vaccine [3,4,5]. Among these, vaccination with dendritic cells (DCs) is an effective approach to overcome the immunosuppressive microenvironment, as well as initiate antitumor-specific immune responses [6,7]. The first step of DC vaccination is maturation of DCs isolated from autologous peripheral blood mononuclear cells (PBMCs) using tumor-associated antigens. Following maturation, DCs are re-administered to the patients to induce antitumor immunity [8]. A successful therapeutic vaccine mostly depends on generating cytotoxic T lymphocytes (CTLs) that specifically recognize and directly kill cancer cells [9]. However, tumor antigens often induce tolerogenicity to host immunity via downregulation of major histocompatibility complex (MHC) molecules to evade recognition and upregulation of immune inhibitory receptors [10]. Due to these complications, immunostimulatory adjuvants are considered a promising strategy to overcome these impediments toward effective cancer vaccines.

Immunostimulatory adjuvants enable solving the problems of immunosuppression and low antigenicity of tumor antigens by inducing a more rapid immune response, such as a strong antibody response and effective T-cell response [11]. The adjuvants can be divided into three types, including delivery systems, immune potentiators, and mucosal adjuvants [12]. Immune potentiators generally target innate immunity signaling pathways by inducing activation and maturation of antigen-presenting cells. Consequently, combination with immune potentiators contributes to long-lasting immunity via increasing the efficacy of DC vaccines [13]. Unfortunately, some side effects of immune potentiators have been suggested (e.g., toxicity, splenomegaly, lymphoid follicle destruction, and immunosuppression) [14]. Therefore, several studies have focused on developing new and safe immune potentiators from natural ingredients [15].

As edible insects are being considered as an attractive protein source, application of these edible insects as medicine or nutrition supplement has been actively sought [16]. *Bombyx batryticatus* (*B. batryticatus*), a popular insect used as traditional medicine in China, originates from the dried stiff larval stage of *Bombyx mori* L. infected by *Beauveria bassiana* [17]. The pharmacological effects of *B. batryticatus* extracts, such as their antibacterial, neuroprotective, and antioxidant activity, have been widely studied [18,19,20]. In particular, recent studies suggested that *B. batryticatus* extracts show strong antitumor activities in various cancer cells, in addition to their potential to regulate the tumor microenvironment via macrophage polarization toward the M1 phenotype, which plays a role in antitumor immunity [17,18,21,22]. *B. batryticatus* contains various nutritional constituents, including proteins, peptides, polysaccharides, flavonoids, fatty acids, and others. Of these, the principal constituents are proteins, which account for 40–70% of total components [23]. Nevertheless, there is no evidence yet supporting the effects of *B. batryticatus* protein extracts on the physiological activities of immune cells.

Here, we investigated whether *B. batryticatus* extracts that are rich in proteins would induce DC maturation in vitro and consequently enhance antigen-specific anticancer immunity in a DC vaccine mouse model.

## 2. Results

### 2.1. Amino-Acid Composition of BBPE

Dietary supplementation with high-quality protein helps to strengthen immune function [24]. Amino-acid composition is an important determinant of the nutritional value of protein. The amino-acid profile of BBPE is presented in Table 1. BBPE contains a balanced content of essential amino acids and nonessential amino acids, with the exception of tryptophan. In BBPE, the content of glutamine and glutamate was highest, followed by alanine, while the histidine content was lowest. In particular, BBPE contains a considerable content of branched-chain amino acids (valine, leucine, and isoleucine), glutamine, and arginine, which are important immuno-nutrients. To investigate whether the nutritional value of BBPE affects the immunological advantage, the phenotypic and functional changes by BBPE in BMDCs were investigated.

### 2.2. B. batryticatus Protein Extract (BBPE) Induces Phenotypic Maturation of BMDCs and Th1 Polarization

Upon maturation, DCs begin to synthetize peptides presented by MHC molecules, along with co-stimulatory molecules, and they produce high levels of proinflammatory cytokines to induce T-cell-mediated immune responses [25]. Prior to investigating the effects of BBPE on DC maturation, cell viability was analyzed to determine noncytotoxic concentrations of BBPE. BBPE treatment was not toxic to BMDCs at concentrations ranging from 10 to 100 μg/mL (Figure 1A,B). We next investigated whether BBPE can induce DC maturation. First, the expression of co-stimulatory molecules and MHC receptors in the presence of BBPE (10, 50, or 100 μg/mL) or lipopolysaccharide (LPS) (100 ng/mL; a positive control for DC maturation) was analyzed by flow cytometry. As shown in Figure 1C, treatment with BBPE significantly increased the expression of CD80, CD86, MHC class I, and MHC class II in a dose-dependent manner. Next, we examined the production of pro- and anti-inflammatory cytokines. BBPE treatment strongly induced the production of Th1-polarizing cytokines, including tumor necrosis factor (TNF-α) and interleukin (IL)-12p70), while it did not affect anti-inflammatory cytokine IL-10 levels (Figure 2A). The finding that BBPE only increased Th1-polarizing cytokine levels was also confirmed by intracellular cytokine staining (Figure 2B). Taken together, these results strongly indicate that BBPE can induce polarization toward the Th1 phenotype, as well as maturation of BMDCs.

### 2.3. BBPE Increases Antigen-Presenting Ability of BMDCs and Reduction of Endocytosis

Mature DCs display incompetent endocytosis, while they generally have strong antigen-presenting ability [26]. To determine the endocytosis capacity of BMDCs after BBPE treatment, the uptake of fluorescein isothiocyanate (FITC)-dextran by BMDCs was examined using flow cytometry. The group treated with 100 μg/mL of BBPE showed a lower percentage of double-positive (CD11c^+^ and dextran-FITC^+^) cells than the control group (Figure 3A). We next investigated the exogenous antigen-presenting ability of BMDCs using flow cytometry. The percentage of cells positive for 25-D1.16, which recognizes MHC class I-associated ovalbumin (OVA)_257–264_, was higher in groups co-treated with OVA protein and 50 μg/mL BBPE than in those treated with OVA only (Figure 3B). Furthermore, the percentage of cells positive for Y-Ae that reacts with Eα_52–68_ peptide (MHC class-II) was higher in groups co-treated with Eα_44–76_ peptide and 50 μg/mL BBPE than in those treated with Eα_44–76_ peptide only (Figure 3C).

### 2.4. BBPE-Treated BMDCs Induce T-Cell Proliferation and Activation

We examined whether BMDCs matured by BBPE could increase T-cell proliferation and activation. Thus, carboxyfluorescein succinimidyl ester (CFSE)-labeled CD4^+^ or CD8^+^ T cells were co-cultured with different groups of DCs prior to performing an allogenic in vitro T cell proliferation assay. As shown in Figure 4A, both CFSE-labeled CD4^+^ and CD8^+^ T cells co-cultured with BBPE-treated DCs showed higher proliferation than the same T cells co-cultured with nontreated DCs. In addition, CD4^+^ or CD8^+^ T cells co-cultured with BBPE-treated DCs produced higher levels of Th1 cytokines (interferon (IFN)-γ and IL-2) than those co-cultured with immature DCs, while Th2 cytokines (IL-5) did not change in CD4^+^ T cells (Figure 4B).

### 2.5. BBPE Improves Antitumor Effects of DC Vaccine via Antigen-Specific Multifunctional T-Cell Activation

A multifunctional T-cell response contributes to orchestrating strong anti-cancer immunity [27]. Therefore, to assess whether BBPE-treated DCs can induce antigen-specific multifunctional T-cell activity in OVA-pulsed DC-immunized mice, splenocytes isolated from each group (PBS, immature DC, OVA-pulsed DC, or OVA/BBPE-pulsed DC-injected) were re-stimulated with OVA peptide, and all possible combinations of three individual parameters (the production of IFN-γ, TNF-α, and IL-2) and CD107a-producing cells were analyzed by flow cytometry. As shown in Figure 5A, the BBPE/OVA-pulsed DC-injected group showed a higher percentage of antigen-specific multifunctional (cells demonstrating ≥ 3 functions; TNF-α^+^IL-2^+^IFN-γ^+^) CD4^+^ T-cell responses (Figure 5A). Additionally, the BBPE/OVA-pulsed DC-injected group showed the increased serum antigen-specific immunoglobulin G2c (IgG2c) response (Th1) without IgG1 response (Th2), which resulted in a higher ratio of IgG2c/IgG1 (Figure 5B). Furthermore, the BBPE/OVA-pulsed DC-injected group showed a higher percentage of antigen-specific multifunctional (TNF-α^+^IL-2^+^IFN-γ^+^, bifunctional (IL-2^+^IFN-γ^+^) CD8^+^ T cells as well as CD107a (cytotoxic granule)-producing CD8^+^ T cell, than OVA-pulsed DC-injected group (Figure 6A). Representative flow cytometry data showing multifunctional, bifunctional, and single-positive T cells are presented in Appendix A. Unstimulated control data are presented in Appendix A. Next, to investigate whether the multifunctional T-cell response induced by BBPE/OVA-pulsed DC immunization can protect against tumor progression, we established an E.G7-OVA intradermal tumor model. After three BBPE/OVA-pulsed DC injections, tumor growth was significantly lower (27 days, 30 days; *p* < 0.05) compared to injection with only OVA-pulsed DC (Figure 6B). Taken together, these results indicate that the BBPE/OVA-pulsed DC vaccine successfully induced and enhanced the Th1-type anti-tumor immunity in the E.G7-OVA intradermal model.

## 3. Discussion

In the current study, to investigate whether BBPE could act as an immune potentiator for the DC vaccine as an effective cancer immunotherapy, we firstly examined the immunological effects of BBPE on phenotypic and functional DC maturation.

DCs, the most powerful antigen-presenting cells (APCs), play pivotal roles in linking innate and adaptive immune response [28]. In their immature state, DCs have specific features, such as high capacity to capture antigens and low expression of co-stimulatory molecules and cytokines. Importantly, DCs activated by various stimuli migrate to lymph nodes and stimulate naïve T cells by expressing high levels of MHC class I/II and co-stimulatory molecules, and producing high amounts of cytokines [29]. Unfortunately, various factors generated by the tumor microenvironment (TME) hamper DC functions and Th1 polarization [30,31]. Therefore, phenotypic maturation of DCs into polarized Th1 is an effective strategy that can yield strong antitumor immunity by enhancing CTL activity, particularly compared to Th2 [32,33]. IL-12 is a key cytokine that enables polarization of CD4^+^ T cells into Th1, in addition to showing inherent antitumor effects [34,35]. Furthermore, the increased ability of DCs to cross-present tumor-associated antigens onto MHC class I is critical for the effective function of CD8^+^ T cells [36].

In the field of clinical nutrition, specific amino acids, such as glutamine, arginine, and branched-chain amino acids (valine, leucine, and isoleucine), are known to mainly regulate the innate immune system [37]. In particular, supplementation with branched-chain amino acids (BCAAs) efficiently leads to maturation and activation of dendritic cells [38,39]. In the present study, we first demonstrated that BBPE contains both essential and nonessential amino acids, including the balanced contents of branched-chain amino acids (Table 1). Subsequently, we found that treatment with BBPE leads to BMDC maturation, including increased co-stimulatory molecule levels, MHC class I and II expression (Figure 1), production of Th1 cytokines (Figure 2), and antigen-presenting ability to MHC class I and II (Figure 3). In this regard, a previous study revealed that the enriched contents of BCAAs in extracellular conditions induce maturation and Th1 polarization of DCs [39]. On this basis, we hypothesized that maturation and Th1 polarization in BMDCs by BBPE are related to the considerable contents of BCAAs in BBPE. Next, we showed that BBPE-treated DCs increased CD4^+^ and CD8^+^ T-cell proliferation and activation (Figure 4). These findings demonstrate that BBPE induces Th1 polarization of BMDCs, as well as maturation. We speculated the protein-rich source in BBPE to be a valuable energy provider for DC maturation and Th1 polarization. Although we used BMDCs after CD11c^+^ selection in loosely adherent cell in all in vitro experiments, some macrophages also highly express CD11c [40]. Therefore, we assumed that BBPE can induce Th1 polarization in professional antigen-presenting cells, including macrophages and DCs.

T cells capable of simultaneously producing two (bifunctional) or three (multifunctional) cytokines, including IFN-γ, IL-2, and TNF-α, have been recently shown to induce a powerful Th1-type immune response, which permits long-lasting antitumor immunity [41]. High-functional T cells producing both IFN-γ and TNF-α can boost local effector cell production in local tissue, which importantly contributes to generating antitumor conditions [27]. Moreover, the high level of IL-2 is related to cytotoxic granules produced by CTL, such as CD107a [42]. In this regard, multifunctional CD4^+^ T cells help to generate an antigen-specific antibody response, as well as CD8^+^ memory T cells [43,44]. CD4^+^ T cells with cytotoxic activity also play an important role in anticancer immunity [45,46]. Therefore, induction of multifunctional CD4^+^ and CD8^+^ T cells is considered an effective strategy for orchestrating anticancer immunity. We used multiparameter flow cytometry to analyze CD4^+^ and CD8^+^ T cells in splenocytes after OVA-DC immunization. In our study, immunization with OVA/BBPE-treated DCs significantly increased the frequency of multifunctional or bifunctional CD4^+^ and CD8^+^ T cells in spleen. Furthermore, OVA/BBPE-treated DC-immunized mice exhibited an enhanced Th1-related antibody response (Figure 5). Collectively, the enhanced OVA-specific Th1-immune response mediated by OVA/BBPE-treated DC immunization resulted in a significant decrease in tumor volume in E.G7 tumor-bearing mice (Figure 6).

## 4. Materials and Methods

### 4.1. Antibodies and Reagents

Recombinant mouse granulocyte-macrophage colony stimulating factor (rmGM-CSF) and rm interleukin-4 (rmIL-4) were purchased from JW CreaGene (Daegu, Korea), whereas 2-mercaptoethanol (2-ME), 4-(2-hydroxyethyl)-1-piperazineethanesulfonic acid (HEPES), and nonessential amino acids were purchased from Invitrogen (Carlsbad, CA, USA). Roswell Park Memorial Institute (RPMI) 1640 medium, fetal bovine serum (FBS), and penicillin/streptomycin were purchased from Gibco (Grand Island, NY, USA). V450-conjugated anti-CD11c, anti-CD107a, V500-conjugated anti-CD8, BV510-conjugated live/dead staining kit, fluorescein isothiocyanate (FITC)-conjugated anti-CD80, anti-I-A-Eα52-68, anti-CD4, anti-IL-10, dextran (40,000 Da), phycoerythrin (PE)-conjugated anti-CD86, anti-H-2kb (SIINFEKL), anti-IL-12p70, anti-IFN-γ, and allophycocyanin (APC)-conjugated anti-TNF-α were purchased from BD Biosciences (San Diego, CA, USA). APC-conjugated anti-MHC-I, APC-Cy7-conjugated anti-CD3, PerCP-Cy5.5-conjugated anti-CD8, and PE-Cy7-conjugated anti-IL-2 and anti-MHC-II were purchased from eBioscience (San Diego, CA, USA). Lipopolysaccharide (LPS) from *Escherichia coli* O111:B4 was purchased from InvivoGen (San Diego, CA, USA).

### 4.2. Preparation of B. batryticatus Proteins Extract

*B. batryticatus* proteins were extracted following the method from Yi et al. [47] with slight modification. First, 100 g of *B. batryticatus* were blended with 600 mL of cold water containing 0.16% ascorbic acid for 1 min using a blender. Then, the obtained insect suspension was sieved through a stainless-steel filter with a pore size of 125 μm. The filtrates were centrifuged at 15,000× *g* for 30 min at 4 °C. From top to bottom, the obtained fractions contained the lipid layer, the supernatant, and the pellet. The thin lipid layer was removed, and the supernatant fraction was collected and freeze-dried. The freeze-dried supernatant (thereafter called *B. batryticatus* protein extract, BBPE) was stored at −20 °C. The yield of the powder extract was 14.93%. After then, the BBPE was resuspended in phosphate-buffered saline (PBS) and filtered using a 0.22 μm syringe filter (Corning, NY, USA). Next, the absence of endotoxin or LPS contamination in BBPE was confirmed using the Limulus Amebocyte Lysate (LAL) assay kit (Lonza, Basel, Switerland) following the manufacturer’s instruction. The content of endotoxin in BBPE was <4 pg/mL (0.1 EU/mL).

### 4.3. Analysis of Amino-Acid Composition

The amino-acid composition of BBPE was measured according to the method described by Bidlingmeyer et al. [48] with slight modification. The BBPE sample was hydrolyzed using 6 M HCl and 1% phenol solution. Hydrolyzed BBPE was taken for precolumn derivatization using phenyl isothiocyanate (PITC). Briefly, 40 μL of hydrolyzed BBPE was mixed with a distilled water (DW)/trethylamine/PITC/MeOH (1:1:1:7) solution for 30 min and dried. The sample was re-dissolved in solvent A (140 mM sodium acetate containing 6% acetonitrile) and filtered using a 0.22 μm filter. Analysis was performed with a Waters 510 HPLC pump using a Waters Pico-tag column (3.9 × 300 mm, 4 μm). HPLC spectra were detected at 254 nm. The separation was performed by the following gradient program: 0 min (0% solvent B), 9 min (14% B), 9.2 min (20% B), 17.5 min (46% B), 17.7 min (100% B), and 21 min (0% B); 60% acetonitrile was used as solvent B.

### 4.4. Cells and Animals

Specific pathogen-free, 6–8 week old female C57BL/6 or BALB/c mice were purchased from Orient Bio, Inc. (Seoul, Korea). All procedures were approved by the Institutional Animal Care and Use Committee of Korea Atomic Energy Research Institute (KAERI-IACUC-2019-001).

Bone marrow-derived dendritic cells (BMDCs) were generated from C57BL/6 mice according to established protocols [49]. Briefly, red blood cells (RBCs) in the whole bone marrow cells were lysed using RBC lysis buffer. Then, 10 mL of RPMI 1640 medium supplemented with 10% FBS, 100 U/mL of penicillin/streptomycin, 20 ng/mL of GM-CSF, 0.5 ng/mL of IL-4, 50 μM 2-mercaptoethanol, 1% minimum essential media (MEM) nonessential amino acids, and 5 mM HEPES buffer (DC culture medium) was added to 1 × 10^6^ cells per 100 mm plate. Next, 10 mL of DC culture medium was added on days 3 and 6 of culture. On day 8, BMDCs were isolated by positive selection using anti-CD11c microbeads (Miltenyi Biotec, Bergisch Gladbach, Germany) in loosely adherent cells.

E.G7 cells, which express ovalbumin (OVA) and are derived from EL4 (thymoma), were purchased from American Type Culture Collection (VA, USA). E.G7 cells were cultured in RPMI 1640 medium supplemented with 2 mM l-glutamine, 4.5 g/L glucose, 10 mM HEPES, 1.0 mM sodium pyruvate, 0.05 mM 2-ME, 0.4 mg/mL G418, 1.5 g/L sodium bicarbonate, 10% FBS, and 100 U/mL of penicillin/streptomycin. BMDCs and E.G7 were cultured at 37 °C in humidified atmosphere containing 5% CO_2_.

### 4.5. Measurement of Cell Viability

Cell viability was determined using the Ez-cytox cell viability (Daeil Lab Service, Seoul, Korea) and Annexin V/propidium iodide (PI) apoptosis detection kit (BD Biosciences). BMDCs were seeded into a 48-well plate (0.25 × 10^6^ cells/well) and treated with LPS (100 ng/mL) or BBPE (1, 10, 50, or 100 μg/mL) for 24 h. Cells treated with PBS were used as a negative control. Cells treated with LPS were used as a positive control. Subsequently, the medium was replaced with 10% Ez-cytox solution for 1 h. The absorbance was measured at 450 nm using a microplate reader (TECAN, Salzburg, Austria). For the annexin V/PI assay, the cells were harvested and washed with PBS and then stained with FITC–annexin V and PI following a previously described method [50]. Cytotoxicity was evaluated by flow cytometry (MACSQuant VYB, Miltenyi Biotec) and analyzed using FlowJo version 10 (TreeStar, Ashland, OR, USA).

### 4.6. Analysis of Surface Molecule Expression in BMDCs

To evaluate the effects of BBPE on DC maturation, cells were stained with specifically labeled fluorescent-conjugated mAbs for 20 min at 25 °C to detect expression of CD80, CD86, MHC class I, and MHC class II in CD11c-positive cells, as previously described [51]. The expression of surface molecules was detected by flow cytometry. LPS (100 ng/mL) was used as a positive control for DC maturation.

### 4.7. Analysis of Cytokines in BMDCs

The levels of TNF-α, IL-12p70, and IL-10 in the cell culture supernatant were measured using commercial enzyme-linked immunosorbent assay (ELISA) kits according to the manufacturer’s instructions (eBioscience).

For intracellular cytokine staining, BMDCs were treated with BBPE or LPS for 12 h in the presence of 1 μg/mL of brefeldin A (GolgiPlugTM, BD Bioscience). The cells were stained with PE-Cy7-conjugated CD11c mAb and the BV510-conjugated live/dead cell staining kit for 20 min at 25 °C. Then, the cells were fixed and permeabilized using a Cytofix/Cytoperm kit following the manufacturer’s instructions (BD Bioscience) and stained with APC-conjugated anti-TNF-α, PE-conjugated IL-12p70, and FITC-conjugated IL-10 mAbs. The intracellular levels of TNF-α, IL-12p70, and IL-10 in CD11c-positive cells were detected by flow cytometry [52].

### 4.8. Analysis of Antigen Uptake

After incubation of BMDCs with BBPE or LPS for 24 h, the cells were pulsed with FITC-conjugated dextran (0.5 mg/mL) for 30 min at 37 °C. The cells were washed with ice-cold PBS three times to stop the reaction and stained with anti-CD11c. The data were collected by flow cytometry.

### 4.9. Analysis of Antigen-Presenting Ability

The BMDCs were stimulated with 500 μg/mL of OVA protein (Sigma-Aldrich, St. Louis, MO, USA) or 25 μg/mL of Eα_44–76_ peptide (RLEEFAKFASFEAQGALANIAVDKANLDVMKKR; underlined sequence binds to MHC-II) in the absence or presence of BBPE for 24 h. To investigate peptide formation of MHC-I or MHC-II, the cells were stained with anti-CD11c, anti-H-2kb (SIINFEKL, eBioscience) or anti-I-Ab-Eα_52–68_ (Y-Ae; ASFEAQGALANIAVDKA, eBioscience) mAbs for 20 min at 25 °C. Data were collected by flow cytometry [52].

### 4.10. Mixed Lymphocyte Reaction (MLR) Assay

To determine the T-cell proliferative ability of BBPE-treated BMDCs, splenocytes were isolated from spleen of BALB/c mice. After RBC lysis, splenocytes were incubated with anti-CD4- or anti-CD8-coated magnetic microbeads (Miltenyi Biotec), and the microbead-conjugated CD4^+^ and CD8^+^ cells were separated using an LS column (Miltenyi Biotec) according to the manufacturer’s instructions. These separated cells were stained with 1 μM CSFE for 20 min at 37 °C and washed with 2% FBS in PBS for 10 min. The BMDCs (0.2 × 10^5^ cells) treated with BBPE or LPS for 24 h were co-cultured with CFSE-stained CD4^+^ and CD8^+^ T cells (1 × 10^5^ cells) in a 96-well U-bottom plate in the presence of 1 μg/mL of plate-bound anti-CD3 and 1 μg/mL of soluble anti-CD28 mAb (eBioscience). After 2 days of co-culture, the T cells were stained with fluorescent conjugated anti-CD4 or anti-CD8 mAb for 20 min and analyzed by flow cytometry. The levels of IFN-γ, IL-5, and IL-2 in cell culture supernatant was measured by ELISA.

### 4.11. Mouse Immunization

To investigate the immune-enhancing capacity of BBPE in DC vaccine mice, the animals were divided into four groups (*n* = 5 per group): (1) PBS, (2) immature DCs, (3) OVA (5 μg/mL of OVA_257–264_ and 5 μg/mL of OVA_323–339_)-pulsed DC, and (4) OVA and 100 μg/mL of BBPE-pulsed DC. The mice of each group were injected intravenously with PBS or each group of DCs (0.1 mL/mouse) on days 1, 3, and 5.

### 4.12. Analysis of Multifunctional T-Cell Subsets in Spleens of Immunized Mice

Two weeks after the last immunization, spleen cells were isolated from each group of mice and lysed with RBC lysis buffer. Next, the cells were re-stimulated with OVA_257–264_ and OVA_323–339_ in the presence of GolgiPlug (0.5 μg/mL), GolgiStop (0.5 μg/mL), and anti-CD107a antibody (2.5 μg/mL, BD Bioscience) for 6 h at 37 °C. Subsequently, the cells were stained with a live/dead cell staining kit, APC-Cy7-conjugated anti-CD3, and V500-conjugated anti-CD8 mAbs for 30 min at 4 °C. After the fixation and permeabilization step, according to the manufacturer’s instructions, cells were stained with PE-conjugated anti-IFN-γ, APC-conjugated anti-TNF-α, and PE-Cy7-conjugated IL-2 mAbs for 30 min at 4 °C. The cells were analyzed using flow cytometry and FlowJo software (BD Bioscience). The intracellular levels of TNF-α, IL-12p70, and IL-10 in CD11c-positive cells were detected by flow cytometry [53].

### 4.13. Measurement of OVA-Specific Antibody

OVA-specific IgG1 (Sigma-Aldrich) and IgG2a (Southern Biotech, Birmingham, AL, USA) antibodies in serum were measured by the indirect ELISA. In brief, plates were coated with OVA_323–339_ peptide (1 μg/mL) and incubated overnight at 4 °C. The wells were washed three times with PBS containing 0.05% (*v*/*v*) Tween-20 (PBS/Tween) and then blocked with PBS containing 5% FBS at 37 °C for 2 h. After washing three times with PBS, sera obtained from each immunized mouse were added (dilution 1:500), and the plates were incubated for 2 h at 37 °C followed by washing three times with PBS. Then, horseradish peroxidase (HRP)-conjugated secondary antibodies against IgG1 and IgG2a were added, and the plates were incubated for 1 h at RT. The reaction was developed using 3,3’, 5,5’’-tetramethylbenzidine (TMB) substrate (Sigma-Aldrich), and the enzyme reaction was terminated by adding 2 N H_2_SO_4_. The optical density was detected at 495 nm within 20 min using a microplate ELISA reader [53].

### 4.14. Tumor Challenge

Mice were injected intradermally into the right lower back with E.G7 cells (2 × 10^6^ cells/mice). PBS, immature DCs (iDCs), DCs pulsed with OVA_257–264_ and OVA_323–339_ (OVA-DCs), or BBPE-treated DCs pulsed with OVA_257–264_ and OVA_323–339_ (BBPE/OVA-DCs) were intravenously injected on days 1, 3, and 5 after tumor implantation. Tumor size was measured every 3 days, and tumor volume was calculated as follows:V = (2A × B)/2,
where A is the length of the short axis and B is the length of the long axis.

### 4.15. Statistical Analysis

All analyses were repeated at least three times with consistent results. The significance for comparisons between samples was determined by one-way ANOVA followed by Tukey’s multiple comparison test or unpaired *t*-tests using statistical software (GraphPad Prism version 5, GraphPad Prism Software, San Diego, CA, USA).

## 5. Conclusions

In the present study, our goal was to evaluate BBPE as an immune potentiator to increase the efficacy of the DC vaccine for cancer immunotherapy. BBPE is mainly composed of 18 amino acids with nine essential ones. BBPE induces BMDC maturation and activation, and it inhibits OVA-expressing tumor growth via enhancing the OVA-specific Th1-immune response in DC vaccine mice. Our results demonstrated that BBPE will be an interesting and effective immune potentiator for the DC-based vaccine for cancer therapy. However, further studies regarding the correlation between the quality control of *B. batryticatus* and nutritional and physiological change are needed for the general use of BBPE as a vaccine adjuvant.

## Figures and Tables

**Figure 1 molecules-26-00476-f001:**
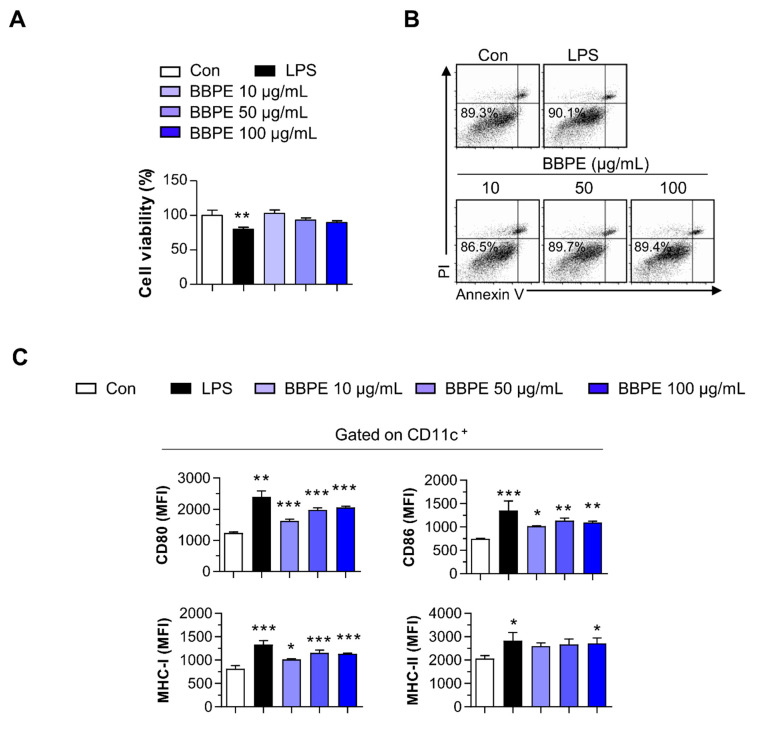
*B. batryticatus* protein extract (BBPE) induces bone marrow-derived dendritic cell (BMDC) maturation without cytotoxicity. BMDCs were treated with BBPE (10, 50, or 100 μg/mL), lipopolysaccharide (LPS, 100 ng/mL), or an equal volume of phosphate-buffered saline (PBS) as a control (Con) for 24 h. Then, cell viability was measured using Ez-cytox cell viability assay kit (**A**) or annexin V/propidum iodide (PI) assay (**B**). BMDCs were stained with anti-CD11c, anti-CD80, anti-CD86, anti- MHC-I, and anti-MHC-II mAbs and analyzed to determine expression of surface molecules (**C**). Bar graphs represent the median fluorescence intensity (MFI) of each surface molecule in CD11c^+^ cells. All bar graphs show the mean ± SD (*n* = 3). Statistical analysis was performed using one-way ANOVA followed by Tukey’s post hoc test. * *p* < 0.05, ** *p* < 0.01, and *** *p* < 0.001 represent significant differences.

**Figure 2 molecules-26-00476-f002:**
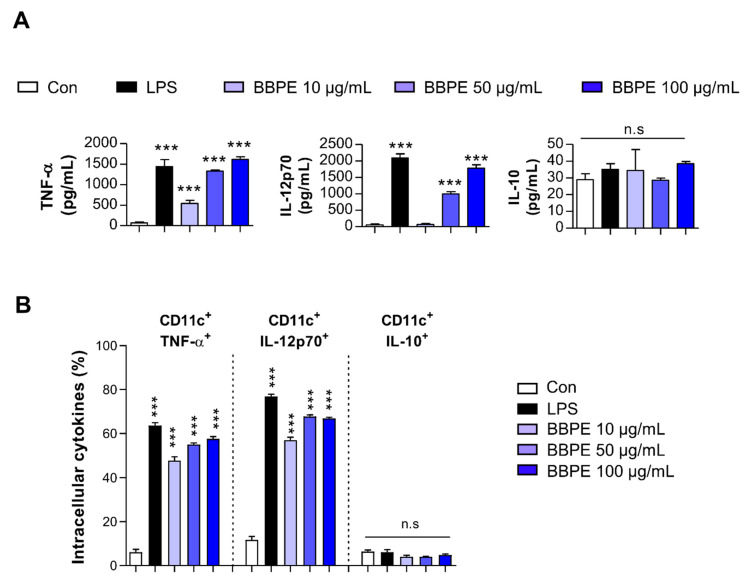
Th1-type cytokine production in BMDCs is increased by BBPE treatment. BMDCs were treated with BBPE (10, 50, or 100 μg/mL), LPS (100 ng/mL), or PBS (Con) for 24 h; then, TNF-α, IL-12p70, and IL-10 levels in culture supernatant were measured by ELISA (**A**). Intracellular levels of TNF-α, IL-12p70, and IL-10 in BMDCs were determined by flow cytometry (**B**). One representative plot of three independent experiments is shown. All bar graphs show the mean ± SD (*n* = 3). Statistical analysis was performed using one-way ANOVA followed by Tukey’s post hoc test. *** *p* < 0.001 represents significant differences.

**Figure 3 molecules-26-00476-f003:**
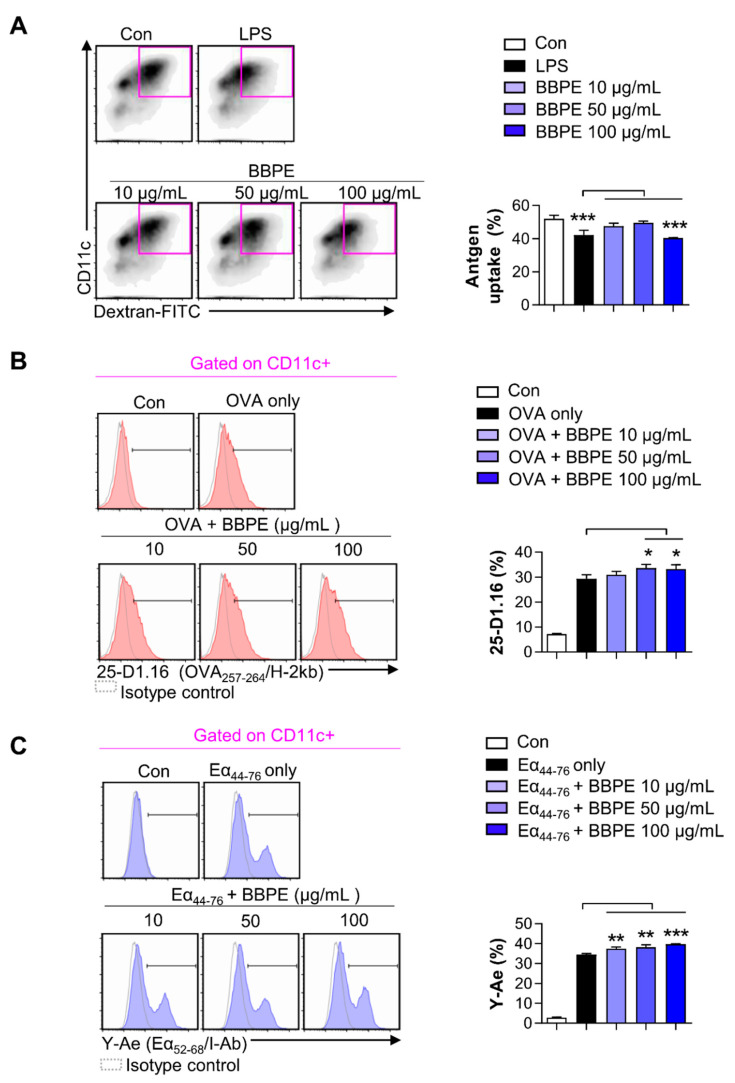
BBPE treatment alters endocytic activity and antigen-presenting ability in BMDCs. The endocytic capacity was assessed by flow cytometry (**A**). One representative plot of three independent experiments is shown. The bar graph represents the percentage of FITC-dextran and CD11c double-positive cells. The antigen-presenting ability of MHC class I (**B**) and MHC class II (**C**) was analyzed by flow cytometry. BMDCs were treated with ovalbumin (OVA) protein (500 μg/mL; **B**) or Eα_44–76_ peptide (50 μg/mL; **C**) in the presence or absence of BBPE (10, 50, or 100 μg/mL) for 24 h, and stained with anti-CD11c, anti-25-D1.16, or anti-Y-Ae mAbs. One representative histogram of three independent experiments is shown. The bar graphs represent the percentage of OVA_257–264_/H-2kb (**B**) or Eα_52–68_/I-Ab (**C**) in CD11c^+^ cells. All bar graphs show the mean ± SD (*n* = 3). Statistical analysis was performed using one-way ANOVA followed by Tukey’s post hoc test. * *p* < 0.05, ** *p* < 0.01, and *** *p* < 0.001 represent significant differences.

**Figure 4 molecules-26-00476-f004:**
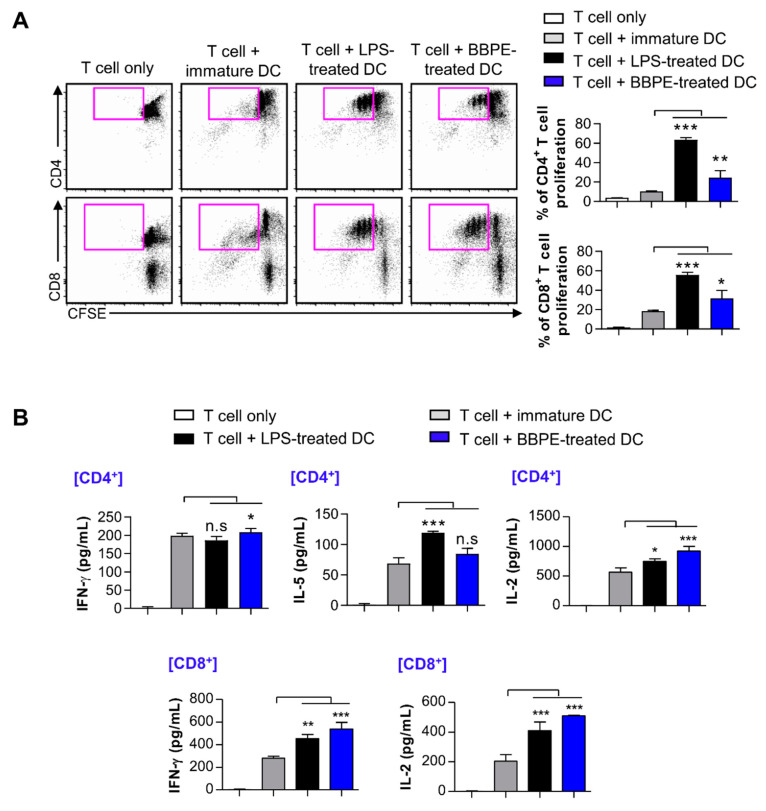
BBPE-treated BMDCs induce proliferation and activation of T cells. DCs were treated with BBPE (100 μg/mL) or LPS (100 ng/mL) for 24 h, and then co-cultured with CFSE-labeled CD4^+^ or CD8^+^ T cells. After 2 days, cells were stained with anti-CD4 or anti-CD8 mAbs, and T-cell proliferation was analyzed by flow cytometry (**A**). IFN-γ, IL-5, and IL-2 levels in the culture supernatant were measured by ELISA (**B**). All bar graphs show the mean ± SD (*n* = 3). Statistical analysis was performed using one-way ANOVA followed by Tukey’s post hoc test. * *p* < 0.05, ** *p* < 0.01, and *** *p* < 0.001 represent significant differences.

**Figure 5 molecules-26-00476-f005:**
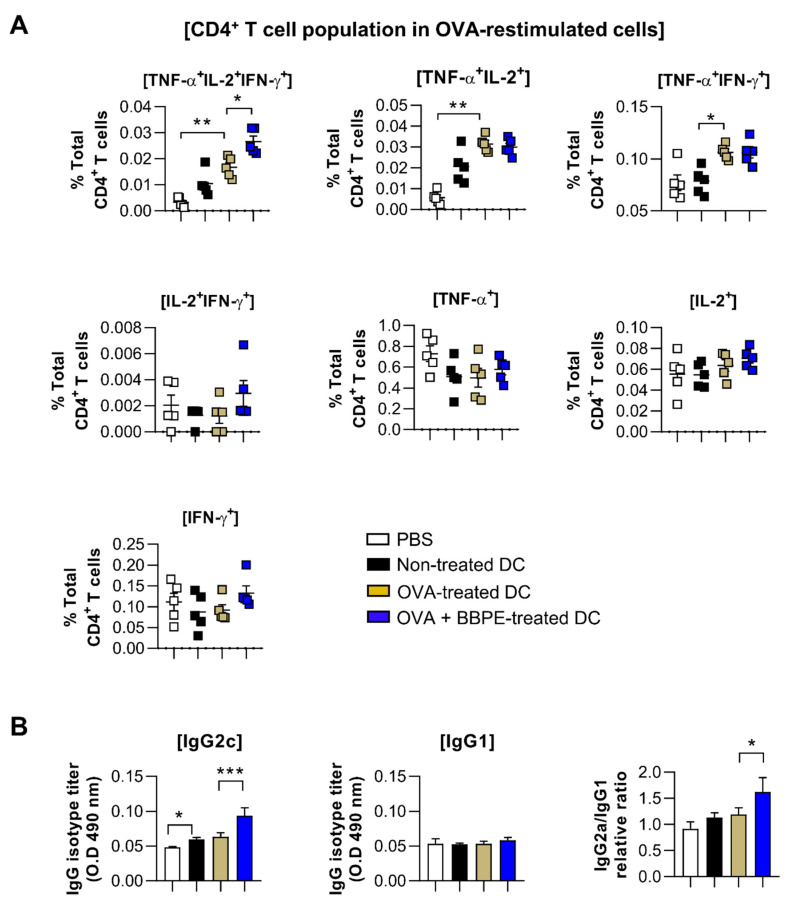
Administration of BBPE/OVA-treated DC vaccine enhances antigen-specific immune responses and antitumor effects. The mice were immunized with PBS, immature DCs, OVA-pulsed DCs, or OVA/BBPE-pulsed DCs (*n* = 5 mice/group). (**A**) After 2 weeks, multifunctional T cells were analyzed by flow cytometry. Each bar graph represents the percentage of antigen-specific multifunctional, bifunctional, or single-positive cells in CD3^+^CD4^+^ T cells. (**B**) OVA_323–339_-specific IgG2c and IgG1 in serum were analyzed by ELISA. All experiments were repeated two times. All graphs show the mean ± SD (*n* = 5) of a representative experiment. Statistical analysis was performed using one-way ANOVA followed by Tukey’s post hoc test. * *p* < 0.05, ** *p* < 0.01 and *** *p* < 0.001 represent significant differences.

**Figure 6 molecules-26-00476-f006:**
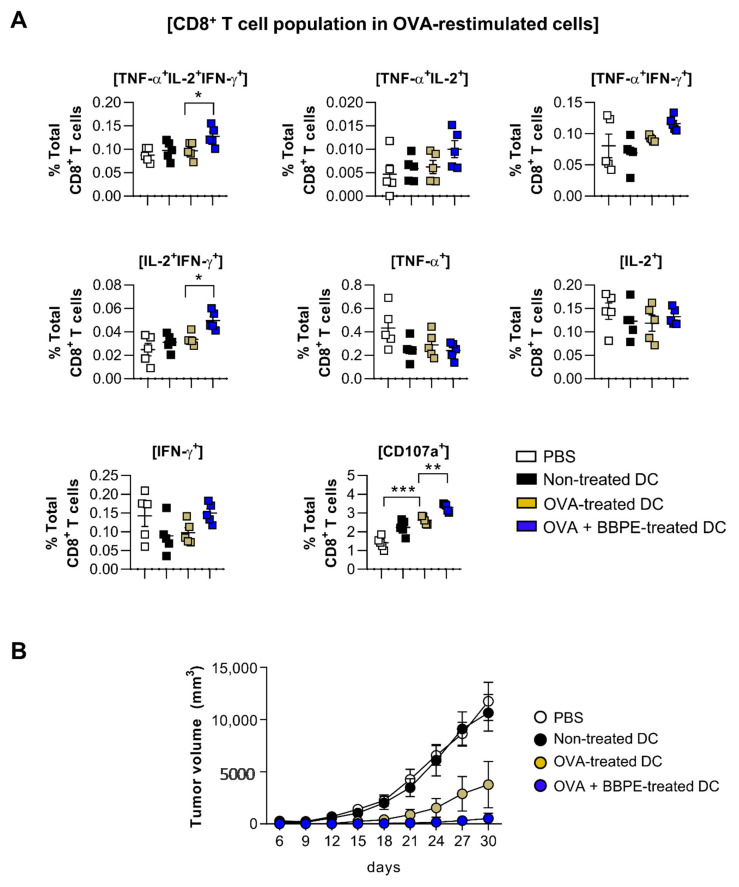
Administration of BBPE/OVA-treated DC vaccine enhances antigen-specific immune responses and anti-tumor effects. The mice were immunized with PBS, immature DCs, OVA-pulsed DCs, or OVA/BBPE-pulsed DCs (*n* = 5 mice/group). (**A**) After 2 weeks, multifunctional T cells were analyzed by flow cytometry. Each bar graph represents the percentage of antigen-specific multifunctional, bifunctional, or single-positive cells in CD3^+^CD8^+^ T cells. (**B**) Mice were intradermally challenged with E.G7 (2 × 10^6^ cells/mouse) and immunized with PBS or each DC group on day 1, 3, and 5, and then the tumor growth was measured every 3 days (*n* = 6 mice/group). All experiments were repeated two times. All graphs show the mean ± SD of a representative experiment. Statistical analysis was performed using one-way ANOVA followed by Tukey’s post hoc test. * *p* < 0.05, ** *p* < 0.01, and *** *p* < 0.001 represent significant differences.

**Table 1 molecules-26-00476-t001:** Amino-acid composition of *Bombyx batryticatus* protein-rich extract (BBPE).

Essential Amino Acid (mg/g)		Nonessential Amino Acid (mg/g)	
Histidine	1.88	Cysteine	2.60
Threonine	11.28	Aspartate + Asparagine	21.18
Valine	16.12	Glutamine + Glutamate	58.89
Methionine	2.82	Serine	17.09
Isoleucine	10.56	Glycine	21.14
Leucine	15.62	Arginine	27.47
Phenylalanine	6.24	Alanine	35.11
Tryptophan	N.D.	Proline	19.22
Lysine	14.51	Tyrosine	5.66
Sum of EAA	79.02	Sum of NEAA	208.37

All amino acid values are expressed as milligram per g of protein. EAA, essential amino acid; NEAA, nonessential amino acid; N.D., not detected.

## Data Availability

Data is contained with the article or Appendix A.

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
