# Peer review of "Bombyx batryticatus Protein-Rich Extract Induces Maturation of Dendritic Cells and Th1 Polarization: A Potential Immunological Adjuvant for Cancer Vaccine"

_molecules, 2021, doi:10.3390/molecules26020476_

Round 1

Reviewer 1 Report

In their study “Bombyx batryticatus protein-rich extract induces maturation of dendritic cells and Th1-polarization: A potential immunological adjuvant for cancer vaccine”, Ha-Yean Song et al. explore the capacity of the protein extract as an adjuvant for cancer-immunotherapy by addressing its ability to stimulate antigen presentation by accessory cells. Given previous findings of positive effects and the use of the protein content in traditional medicine, unraveling the putative mode of action of the compounds is interesting and has the potential to be of relevance. In a well written and easy-to-follow manuscript, the authors present compelling evidence that the protein extract indeed stimulates antigen presentation in a Th1-biasing manner. Some clarifications and important controls are however lacking in the current form of the manuscript.

Specific points:

  • Can the authors exclude LPS contamination in their protein extracts? As this could potentially explain the entire phenotype, this is essential. Relevant KO models could be used instead to address this question.
  • The authors use GM-CSF cultures of BM derived cells and refer to those as dendritic cells. These cultures are however known to be highly heterogenous, with most cells being macrophages. The term “dendritic cells” should therefore be omitted, unless the authors are able to show data from bona fide DCs (ex vivo or FLT3L cultures)
  • Statistical analysis: Tukey is the post hoc analysis, the actual test should be ANOVA. Why did the authors switch to unpaired t-tests in figure 5 and 6?
  • The number of samples per experiment as well as the number of experimental repeats should be reported for each figure.
  • In Figure 5B, the authors report antibody titers. If this experiment was performed only once, it should be repeated. It is not immediately clear from the results or the figure legend, but the material and method section suggests that this is a C57Bl/6 experiments. Mice on this background do not have IgG2a.
  • Gating strategies including an unstimulated control or isotype control should be provided for the data presented in Figures 5 and 6. Can the authors provide supplementary evidence that the cells in gate “B” are indeed all TNFa+?

Author Response

In their study “Bombyx batryticatus protein-rich extract induces maturation of dendritic cells and Th1-polarization: A potential immunological adjuvant for cancer vaccine”, Ha-Yean Song et al. explore the capacity of the protein extract as an adjuvant for cancer-immunotherapy by addressing its ability to stimulate antigen presentation by accessory cells. Given previous findings of positive effects and the use of the protein content in traditional medicine, unraveling the putative mode of action of the compounds is interesting and has the potential to be of relevance. In a well written and easy-to-follow manuscript, the authors present compelling evidence that the protein extract indeed stimulates antigen presentation in a Th1-biasing manner. Some clarifications and important controls are however lacking in the current form of the manuscript.

  • The authors would like to express a sincere appreciation to your kind review, and will try our best to make adequate corrections based on your comments.

Specific points:                                                 

  • Can the authors exclude LPS contamination in their protein extracts? As this could potentially explain the entire phenotype, this is essential. Relevant KO models could be used instead to address this question.
  • Answer: We would like to thank the reviewer for important comments. To determine whether BBPE is contaminated by LPS, we analyzed the amount of endotoxin in BBPE using the Limulus Amebocyte Lysate (LAL) assay. The endotoxin content in BBPE was < 4 pg/mL (0.1 EU/mL). Thus, we believe that DC maturation by BBPE treatment is not due to contaminating endotoxins or LPS. This information is added in method section (Line 298-302).
  • The authors use GM-CSF cultures of BM derived cells and refer to those as dendritic cells. These cultures are however known to be highly heterogenous, with most cells being macrophages. The term “dendritic cells” should therefore be omitted, unless the authors are able to show data from bona fide DCs (ex vivo or FLT3L cultures)
  • Answer: Thank you for your reasonable comments. Some researchers recommend discarding highly adherent cells and sorting CD11c+ cells in loosely adherent cell to use BMDC. Because CD11c+ macrophage is generally existent in highly adherent cells. Therefore, we used BMDC after magnetic CD11c+ selection in loosely adherent on day 8 culture. This information was added in method section (Line 328-330). However, as you suggested, we will use CD11c+ cells in F4/80-deleted cells for high purity BMDC in our next experiments.
  • Statistical analysis: Tukey is the post hoc analysis, the actual test should be ANOVA. Why did the authors switch to unpaired t-tests in figure 5 and 6?
  • Answer: Thank you for your correct comments. The authors revised the statistical method to one-way ANOVA followed by Tukey’s post hoc test in figure 5 and 6. Furthermore, the sentence regarding statistical method was revised to more accurate in caption of figure 1-4. The revised parts were marked with red.
  • The number of samples per experiment as well as the number of experimental repeats should be reported for each figure.
  • Answer: As you suggested, we marked number of experimental repeats in each figure caption.
  • In Figure 5B, the authors report antibody titers. If this experiment was performed only once, it should be repeated. It is not immediately clear from the results or the figure legend, but the material and method section suggests that this is a C57Bl/6 experiments. Mice on this background do not have IgG2a.
  • Answer: We would like to express a sincere appreciation to your essential comments. Many researchers measure IgG2c instead of IgG2a for Th1 immune response in serum isolated from C57BL/6 mice [1]. Based on this, we revised antibody titer data that measures OVA-specific IgG2c instead of IgG2a in mouse serum. Revised words in manuscripts were marked with red.
  • [1] Nazeri, S.; Zakeri, S.; Mehrizi, A. A.; Sardari, S.; Djadid, N. D. Measuring of IgG2c isotype instead of IgG2a in immunized C57BL/6 mice with Plasmodium vivax TRAP as a subunit vaccine candidate in order to correct interpretation of Th1 versus Th2 immune response. Exp Parasitol 2020, 216, 107944, doi.org/10.1016/j.exppara.2020.107944.
  • Gating strategies including an unstimulated control or isotype control should be provided for the data presented in Figures 5 and 6. Can the authors provide supplementary evidence that the cells in gate “B” are indeed all TNFa+?
  • Answer: As you suggested, we revised gating strategy including unstimulated control in supplementary Figure 4. Non-specific CD107a production was observed in unstimulated cells. We think that this result can be due to anti-CD107 antibody is incubated for 6 h with OVA stimulation (or without stimulation) described as method section. Therefore, we revised that triple cytokine positive cells (TNF-α+IL-2+IFN-γ+) called as multifunctional T cells except CD107a+ cells. Single CD107a+ cells are separately considered and presented in Fig. 6. The percent population of multifunctional-, bifunctional T cells, and single positive cells in unstimulated group is presented in figure S1. Additionally, we believe that reviewer can be better understanding for gating strategy by figure S1. The graph of multi-, bifunctional T cells, or single positive T cells in unstimulated cells was presented in figure S2. All revised parts were marked with red (Line 182-183, 188, 190-194).

Reviewer 2 Report

In this paper, the immunomodulatory role of B. batryticatus protein-rich extract in bone marrow-derived dendritic cells and DC vaccine-immunized mice was investigated. Several other studies have reported the anti-cancer activity of B. batryticatus extracts, but their capacity as immune potentiators for increasing the efficacy of cancer immunotherapy, has not been studied, therefore, from this point of view it is a contribution. In general, the work is well written, but there are things that can be improved.

Results:

How important is it that tryptophan was not detected in the evaluation of the amino acid composition of BBPE?

The figures are very ornate. They contain too much information, making analysis difficult. They need to be simplified. I think it is not necessary to show a representative graph or histogram of 3 or 5 different experiments. With bar graphs or a summary table, it is enough to show the results obtained, without reloading information.

Materials and Methods:

In Materials and Methods, scientific names such as B. batryticatus and Escherichia coli, must be italicized or bold.

In Materials and Methods, the tests described in most the items, require a bibliography to support them.

It remains to justify why PBS or LPS were used as control.

Why the treatments were in a time of 24 hrs?

Need to argue, why IL-5, TNF-α, IL-12p70, IL-10, IFN-γ and IL-2, among other, were the cytokines selected to be evaluated.

Discussion:

In the discussion of the results, it is necessary to connect the evaluation of the amino acid content with the results as immune potentiator for DC-based vaccine in anti-cancer therapy.

It remains to be discussed if the amino acid composition is constant between samples of different insects, does the composition change with age, size, harvest time? How is it possible to standardize the results of extracts obtained from different groups of insects?. If the composition of the extract changes, the conclusions are valid only for the extract studied in this paper, but it is not possible to generalize.

Author Response

In this paper, the immunomodulatory role of B. batryticatus protein-rich extract in bone marrow-derived dendritic cells and DC vaccine-immunized mice was investigated. Several other studies have reported the anti-cancer activity of B. batryticatus extracts, but their capacity as immune potentiators for increasing the efficacy of cancer immunotherapy, has not been studied, therefore, from this point of view it is a contribution. In general, the work is well written, but there are things that can be improved.

  • The authors really appreciate your review of our manuscript, and dedicated our best effort to revising the manuscript depending on your comments.

Results:

How important is it that tryptophan was not detected in the evaluation of the amino acid composition of BBPE?

  • Answer: Thank you for your helpful comment. Previous study revealed that tryptophan and its metabolites can suppress certain immune cells via inducing tolerance [1]. Immune tolerance generally considered as problem in cancer microenvironment. Therefore, it is advantageous for possibility as adjuvant that tryptophan did not detect in BBPE.
  • [1] Moffett, J. R., & Namboodiri, M. A. (2003). Tryptophan and the immune response. Immunology and cell biology, 81(4), 247-265.

The figures are very ornate. They contain too much information, making analysis difficult. They need to be simplified. I think it is not necessary to show a representative graph or histogram of 3 or 5 different experiments. With bar graphs or a summary table, it is enough to show the results obtained, without reloading information.

  • Answer: As you suggested, we removed the representative histogram or dot plot in figure 1,2,5, and 6. Instead, we attached representative dot plot data for better understanding gating strategy of multifunctional T cells in figure S1. We think that representative histogram or dot plot data in figure 3 and 4 are need for better understanding of experiments.

Materials and Methods:

In Materials and Methods, scientific names such as B. batryticatus and Escherichia coli, must be italicized or bold.

  • Answer: Following your suggestion, the scientific names, including B. batryticatus and Escherichia coli, are italicized.

In Materials and Methods, the tests described in most the items, require a bibliography to support them.

  • Answer: As you suggested, we added the bibliography that referring each experimental method. The added references are marked with red.

It remains to justify why PBS or LPS were used as control.

  • Answer: Thank you for your reasonable comment. PBS was used as negative control in because freeze dried BBPE was resuspended in PBS, and then BMDCs were treated with filtered BBPE solution. LPS, known as a TLR4 agonist, is generally used as positive control in in-vitro experiments because it strongly induces DC maturation and cytokine production. inducing strong inflammatory response [2, 3]. Simplified sentences regarding above information were added in Line 298-299 and 342-343.
  • [2] Kwon, D. H., Lee, H., Park, C., Hong, S. H., Hong, S. H., Kim, G. Y., ... & Choi, Y. H. (2019). Glutathione induced immune-stimulatory activity by promoting M1-like macrophages polarization via potential ROS scavenging capacity. Antioxidants, 8(9), 413.
  • [3] Monmai, C., Go, S. H., Shin, I. S., You, S., Lee, H., Kang, S., & Park, W. J. (2018). Immune Enhancement Effect of Asterias amurensis Fatty Acids through NF-κB and MAPK Pathways on RAW 264.7 Cells. Journal of microbiology and biotechnology, 28(3), 349-356.

Why the treatments were in a time of 24 hrs?

  • Answer: The point that you have requested us was very important to understanding our study. Previous study revealed that IL-12 and IL-10 were detected in BMDC supernatant after 5 h LPS stimulation, subsequently, IL-12 production was rising up to 22 h [4]. Furthermore, many researchers measure cytokine in BMDC culture supernatant after 24 h stimulation. Therefore, we measured the markers of DC maturation (surface molecules, cytokine, Ag uptake/presentation) after 24 h.
  • [4] Jiang, H. R., Muckersie, E., Robertson, M., Xu, H., Liversidge, J., & Forrester, J. V. (2002). Secretion of interleukin‐10 or interleukin‐12 by LPS‐activated dendritic cells is critically dependent on time of stimulus relative to initiation of purified DC culture. Journal of Leukocyte Biology, 72(5), 978-985.

Need to argue, why IL-5, TNF-α, IL-12p70, IL-10, IFN-γ and IL-2, among other, were the cytokines selected to be evaluated.

  • Answer: TNF-α and IL-12p70 are most important cytokine for Th1-polarization in DCs [5]. By contrast, IL-10 is well-known anti-inflammatory cytokine. Therefore, to investigate whether BBPE induce Th1 polarization in BMDC, we measured TNF-α, IL-12p70, and IL-10 in BMDC culture supernatant. TNF-α, IL-2 and IFN-γ are considered as most important Th1 cytokines produced by T cells. Among them, IL-2 is crucially involved in T cell proliferation [6]. IL-5 is typical Th2 cytokine produced by T cells. Therefore, to examine BBPE-treated DC induce Th1 polarization of CD4 T cell and CTL activation (Th1), we measured IFN-γ (Th1), IL-5 (Th2), and IL-2 (Th1 & proliferation) in culture supernatant obtained by MLR assay. In in-vitro results, we found that BBPE strongly induced Th1 polarization in BMDC as well as T cells co-cultured with BBPE-treated BMDCs. Additionally, to evaluate whether antigen-specific-Th1 immune response in DC-vaccine mice model, we measured antigen-specific TNF- α, IL-2, and IFN- γ (called as multifunctional T cells) in splenocyte.
  • [5] Feili‐Hariri, M., Falkner, D. H., & Morel, P. A. (2005). Polarization of naive T cells into Th1 or Th2 by distinct cytokine‐driven murine dendritic cell populations: implications for immunotherapy. Journal of leukocyte biology, 78(3), 656-664.
  • [6] Kisuya, J., Chemtai, A., Raballah, E., Keter, A., & Ouma, C. (2019). The diagnostic accuracy of Th1 (IFN-γ, TNF-α, and IL-2) and Th2 (IL-4, IL-6 and IL-10) cytokines response in AFB microscopy smear negative PTB-HIV co-infected patients. Scientific reports, 9(1), 1-12.

Discussion:

In the discussion of the results, it is necessary to connect the evaluation of the amino acid content with the results as immune potentiator for DC-based vaccine in anti-cancer therapy.

  • Answer: The reasonable points that you have requested us were truly essential to improving the quality of this study. In DC-based vaccine therapy, regulation of immune response into Th1 polarization is most important factor. Previous studies demonstrated that branched chain amino acids (BCAAs), including valine, leucine, and isoleucine, play important role in inducing Th1 immune response, specifically, enrich BCAAs condition induce maturation and Th1 polarization. Based on this, we hypothesized that maturation and Th1 polarization in BMDCs by BBPE are related to considerable contents of BCAAs in BBPE. We added these sentences and related bibliography are added in discussion section (Line 245-248).

It remains to be discussed if the amino acid composition is constant between samples of different insects, does the composition change with age, size, harvest time? How is it possible to standardize the results of extracts obtained from different groups of insects?. If the composition of the extract changes, the conclusions are valid only for the extract studied in this paper, but it is not possible to generalize.

  • Answer: Thank you for your accurate comments. B. batryticatus is made by artificial inoculation of Beauveria bassiana to silkworms. Several studies regarding quality control of B. batryticatus are initiated in recent years. Xing et al. demonstrated that stiff development stage is a major determinant of chemical components of B. batryticatus [7]. Therefore, we believe that quality control of B. batryticatus might be more simple than other insects.
  • Our present study is the first research that can suggest possibility of B. batryticatus extract as adjuvant for cancer immunotherapy. However, further studies about correlation between stiff stage and nutritional/physiological change must be required to generalize. We added this limitation in conclusion (Line 454-456).
  • [7] Xing, D., Shen, G., Li, Q., Xiao, Y., Yang, Q., & Xia, Q. (2019). Quality Formation Mechanism of Stiff Silkworm, Bombyx batryticatus Using UPLC-Q-TOF-MS-Based Metabolomics. Molecules, 24(20), 3780.

Round 2

Reviewer 1 Report

The authors addressed my original points to varying degrees. While the manuscript approved, some points of concern remain:

1 - Despite the explanations given by the authors, it would be more accurate to call the cells in question “mononuclear phagocytes” or “antigen-presenting cells” instead of dendritic cells. At the very least, this point should be added to the discussion.

2 - The legends to figure 1, 5 and 6 continue to lack a statement about number of repeats. If done only once, a repeat is required for all data shown.

3 - In the B panel of Figure 5, one of the IgG2c labels should probably say IgG1?

Author Response

The authors addressed my original points to varying degrees. While the manuscript approved, some points of concern remain:

  1. Despite the explanations given by the authors, it would be more accurate to call the cells in question “mononuclear phagocytes” or “antigen-presenting cells” instead of dendritic cells. At the very least, this point should be added to the discussion.
  • Answer: Thank you for your precise comments. Following your suggestion, we added the sentences “Although we used BMDCs after CD11c positive selection in loosely adherent cell in- all in-vitro experiments, some macrophages also highly express CD11c. Therefore, we assumed BBPE can induce Th1 polarization in professional antigen-presenting cells, including macrophage as well as DC.” in discussion section (Line 253-256).
  1. The legends to figure 1, 5 and 6 continue to lack a statement about number of repeats. If done only once, a repeat is required for all data shown.
  • Answer: The in-vivo experiment (figure 5 and 6) was repeated two times. All graphs show mean ± SD (n = 5 or 6) of a representative experiment. We added this information in figure captions and marked with red.
  1. In the B panel of Figure 5, one of the IgG2c labels should probably say IgG1?
  • Thank you for your helpful comment. We revised the one of IgG2c to IgG1.

Reviewer 2 Report

The authors have responded satisfactorily to my comments, so I consider that the paper can be published in the current version..

Author Response

Comments and Suggestions for Authors

The authors have responded satisfactorily to my comments, so I consider that the paper can be published in the current version.

  • The authors would like to express a sincere appreciation to your decision.